# IMPLICIT NEURAL VIDEO COMPRESSION

**Yunfan Zhang    Ties van Rozendaal    Johann Brehmer    Markus Nagel    Taco S. Cohen**
Qualcomm AI Research*
{yunfzhan, ties, jbrehmer, markusn, tacos}@qti.qualcomm.com

## ABSTRACT

We propose a method to compress full-resolution video sequences with implicit neural representations. Each frame is represented as a neural network that maps coordinate positions to pixel values. We use a separate implicit network to modulate the coordinate inputs, which enables efficient motion compensation between frames. Together with a small residual network, this allows us to efficiently compress P-frames relative to the previous frame. We further lower the bitrate by storing the network weights with learned integer quantization. Our method, which we call *implicit pixel flow* (IPF), offers several simplifications over established neural video codecs: it does not require the receiver to have access to a pretrained neural network, does not use expensive interpolation-based warping operations, and does not require a separate training dataset.

## 1   INTRODUCTION

Video streaming makes up a major portion of today's internet traffic. Compression codecs based on deep learning (Lu et al., 2019; Agustsson et al., 2020) have recently become competitive with popular classical codecs like H.264 (AVC) Wiegand et al. (2003) and H.265 (HEVC) Sullivan et al. (2012), but these methods have not yet been widely adapted in real-life applications. We argue one reason lies in practicality: neural codecs require access to a (typically large) neural network on each device on which videos need to be decompressed. This is memory-heavy, difficult to maintain, and can be vulnerable to corruption. Moreover, standard neural codecs require a training dataset that is similar to the video samples expected at test time; the compression performance potentially suffers from training set bias and domain shift.

We propose *implicit pixel flow* (IPF), a method for video compression based on implicit neural representations (INR) that addresses these practical shortcomings. Each frame is represented as a function (modeled as a neural network) that maps coordinates within the frame to RGB values. Encoding then consists of overfitting the network weights on the video frames. Decoding only requires forward passes of the network. We quantize the neural network weights with fixed-point integer quantization with learned parameters and separate per-channel bit widths.

To further reduce the bitrate for video data, we compress most frames as P-frames, i. e. using the information from the previous frame. We leverage the similarity of subsequent frames through a separate implicit network that outputs the optical flow field. We argue that implicit neural representations are a natural fit for such an optical flow warping operation: they require a simple addition in the input space, avoiding the usual interpolation-based operations that are difficult to implement on device Lu et al. (2019); Agustsson et al. (2020). In addition to the lightweight flow network, we train an equally lightweight residual network to complete the modeling of a P-frame.

## 2   RELATED WORK

**Implicit neural representations**    Implicit representations have been successfully used for learning three-dimensional structures Mescheder et al. (2019); Chen & Zhang (2019); Deng et al. (2020); Park et al. (2019); Atzmon & Lipman (2020); Genova et al. (2019; 2020); Jiang et al. (2020) and light fields (see Yariv et al. (2020); Mildenhall et al. (2020); Niemeyer et al. (2020); Park et al. (2021); Liu

---

*Qualcomm AI Research is an initiative of Qualcomm Technologies, Inc.

et al. (2019; 2020); Li et al. (2021); Sitzmann et al. (2019) and references therein). Similar to our approach to compression, these works train a neural network on a single scene such that it is encoded by the network weights.

**Instance-adaptive compression**   Recently, Rozendaal et al. (2021); van Rozendaal et al. (2021) introduced instance-adaptive fine-tuning, in which a compressive autoencoder model is fine-tuned on each test instance. The network weight updates are entropy-coded to the bitstream and transmitted alongside the latents. While this approach relaxes the requirements on the model to generalize from the training dataset to any instance encountered at test time, it still requires a pretrained model to be available for decoding.

**Neural implicit compression codecs**   The first publication to apply implicit neural representation to compression is Dupont et al. (2021). The authors propose to compress images using SIREN-based models (Sitzmann et al., 2020) with varying numbers of layers and channels and quantize them to 16-bit precision. Concurrently to this work, Strümpler et al. (2021) proposed another image codec. It achieves a good compression performance, but it does so by storing a meta-learned model at receiving end, giving up some of the practical advantages of INR codecs as explained in Sec. 1. Recently, Chen et al. (2021) introduced an INR compression codec for video. While performant, this model requires (de)compressing the entire video at once, and is not suitable for streaming applications.

**Dynamic scene representations**   Implicit representations are continuous in nature. Shifting the input to these networks corresponds to continuous spatial translations within the represented scene or image. The closest related work we are aware of is Park et al. (2021), which introduces an auxiliary network to model dynamic warping between smartphone selfies. Unlike selfies, videos frames are inherently sequential. We take advantage of this natural ordering in designing our method.

## 3   METHODS

**Overview**   Encoding a video consists of training a network with quantized weights, minimizing the rate-distortion loss

$$L_{\text{IPF}}(\theta, \tau, \omega) = \underbrace{\mathbb{E}_{t,x,y}\|f_{Q_{\tau_t}(\theta_t)}(x,y) - I_{t,x,y}\|_2^2}_{D} + \beta \underbrace{\mathbb{E}_t \sum_i b_{i,t}}_{R} . \tag{1}$$

Here $t$ is the frame index, $x, y$ are the coordinates within a video frame and $I_{t,x,y}$ are the ground-truth RGB values at these coordinates. $f_{\theta_t}(x,y)$ is the implicit neural network with weights $\theta_t$ evaluated at coordinates $(x, y)$. $Q_{\tau_t}$ is the quantization function with parameters $\tau_t$, and $b_{i,t}$ are the learned bit-widths of the parameters, a function of quantization parameters $s_i$ and $\theta_{\max i}$ (to be defined below).

**Implicit image representations**   At the heart of our compression codec is the choice of neural network architecture used to represent individual images. We base our design on SIREN (Sitzmann et al., 2020). While expressive, SIREN requires one forward pass for each pixel in decoding an image, which is expensive on full resolution media. To lighten the compute, we share part of the computation between neighboring pixels. We implement the MLP as 1x1 convolution layers, between which we insert bilinear interpolation layers to perform upsampling. This reduction in compute comes at the cost of a reduced expressivity in our motion compensation scheme. We therefore restrict to a single upsampling layer with stride 2. This makes both the forward and backwards pass three times faster, while maintaining a good compression performance.

**Implicit video representations**   Video data often have strong redundancies between subsequent frames. Neural implicit representations can represent video data by extending the input space by a third time or frame dimension (Xian et al., 2021; Mehta et al., 2021). While this approach is straightforward, we find that the implicit networks are not expressive enough to represent high-resolution video data at low distortion. While Chen et al. (2021) overcomes this by introducing upsampling similar to our architecture, it still requires encoding the entire video at once and is not suitable for streaming operations.

Instead, we propose to compress video sequences frame by frame while still leveraging the similarity between them. We split the video in small blocks of frames ("groups of pictures" or GoP). The first frame in a GoP is compressed as an I-frame, training a single network to compress an image. The remaining frames in the GoP are trained as P-frames, i. e. using the previous frame as reference. We compress each P-frame with separate flow and residual models that model its change with respect to the previous frame.

In this work we model optical flow implicitly by leveraging the fact that implicit representations are continuous. Recall that frames are represented as a network that takes image coordinates as input, $(x, y) \rightarrow f_t(x, y) = f_{\theta_t}(x, y) = (r, g, b)$. Applying the displacement from an optical flow field $h_\phi(x, y) = (\Delta_t^x, \Delta_t^y)$ requires only to add the displacement vector to the input variables:

$$(x, y) \rightarrow f_t(x, y) = f_{t-1} \circ (1 + h_{\phi_t})(x, y)$$
$$= f_{t-1}(x + \Delta_t^x, y + \Delta_t^y) \,. \tag{2}$$

The displacement fields $(\Delta_t^x, \Delta_t^y)(x, y)$ are represented implicitly as neural networks $h_{\phi_t}$ with weights $\phi_t$, using smaller SIREN architectures (see above).

On top of the warped frames, we model residuals with a separate implicit network $r_{\psi_t}(x, y)$.

$$(x, y) \rightarrow f_t(x, y) = f_{t-1} \circ (1 + h_{\phi_t})(x, y) + r_{\psi_t}(x, y) \,, \tag{3}$$

We find using same sized models for flow and residual allows for the best rate-distortion trade off.

**Quantization and entropy coding**  To reduce the model size of the implicit models representing I-frames, optical flow, and residuals, we quantize every weight tensor $\theta^{(l)} \in \theta$ using fixed-point representation. To learn the quantization parameters and bit-width jointly with the model weights, we follow the parameterization suggested by Uhlich et al. (2020) and learn the scale $s$ and the clipping threshold $\theta_{\max}$. The bit-width $b$ is then implicitly defined as

$$b(s, \theta_{\max}) = \log_2 \left( \left\lceil \frac{\theta_{\max}}{s} \right\rceil + 1 \right) + 1 \,. \tag{4}$$

Uhlich et al. (2020) showed that this parameterization is favorable over learning the bit-width directly as it does not suffer from an unbounded gradient norm. We further extend this approach to per-channel quantization (Krishnamoorthi, 2018) allowing us to learn a separate range and bit-width for every row in the matrix (e.g. Output channel in case of a convolutional layer). Our per-channel mixed precision quantization function is defined as:

$$Q_\tau(\theta_{ij}) = \begin{cases} s_i \cdot \left\lfloor \frac{\theta_{ij}}{s_i} \right\rceil & |\theta_{ij}| \leq \theta_{\max,i}, \\ \text{sign}(\theta_{ij}) \cdot \theta_{\max,i} & |\theta_{ij}| > \theta_{\max,i}. \end{cases} \tag{5}$$

Next, we encode all quantization parameters $\tau = \{s^{(l)}, b^{(l)}\}_{l=1}^L$ and all integer tensors $\theta_{\text{int}} = \{\theta_{\text{int}}^{(l)}\}_{l=1}^L$ to the bitstream. The $s^{(l)}$ are encoded as 32-bit floating point vectors, the bit-widths $b^{(l)}$ as 5-bit integer vectors, and the $\theta_{\text{int}}^{(l)}$ in their respective per-channel bit-width $b_i^{(l)}$.

## 4 EXPERIMENTS

We compress the 7 videos from the UVG-1k dataset (Mercat et al., 2020), which have a Full-HD resolution ($1920 \times 1080$ pixels). As the content and style of these sequences do not change over the video, we only use the first 300 frames of each sequence [1] to save computational resources. We use three different architectures, corresponding to different working points on the rate-distortion curve. We specify the I-frame codec sizes to roughly cover the range of bit-rates we are interested in. For each model, the flow and residual models are then specified to be $1/40$ the size of the I-frame codec, a ratio we empirically determined to give good rate-distortion performance.

We show reconstructions of the first GoP from the Bosphorus sequence in Fig. 1. As baselines we compare to the popular classical codecs mpeg4-2 (ISO, 2004), H.264 (Wiegand et al., 2003), and

---

[1]This is the full sequence for one of the videos and half of the video for the remaining 6.

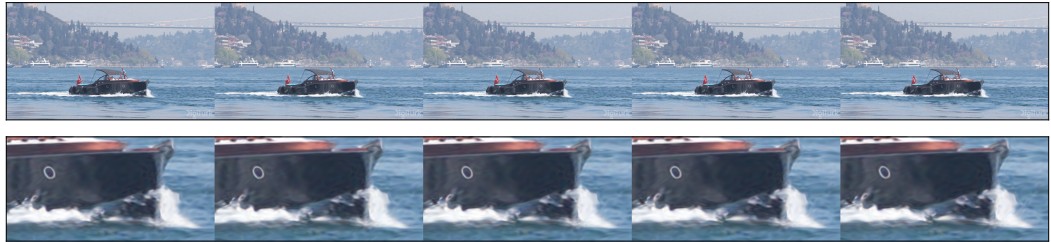

Figure 1: (Top) Reconstruction of the first GoP of the Bosphorus video using our medium sized model. (Bottom) the same reconstructions zoomed in towards from of the boat.

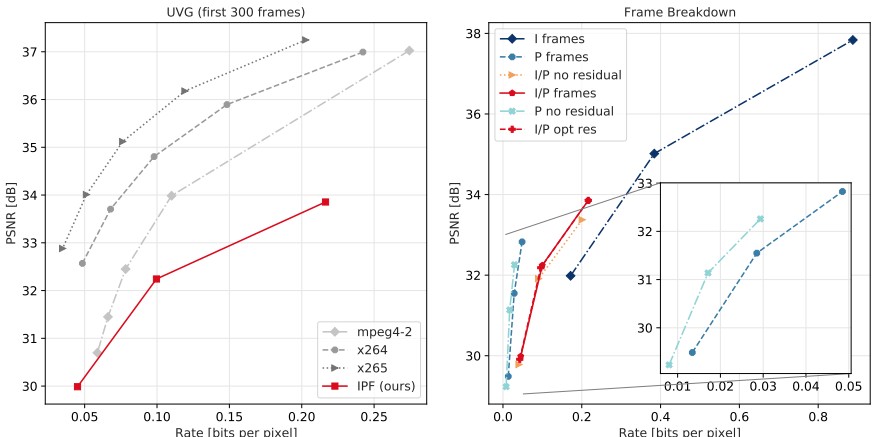

Figure 2: (Left) Overall video compression performance of IPF and three baselines on the UVG dataset. Due to computational constraints we only show result of the first 300 frames. (Right) Breakdown of performance for various frame types. The overall performance (from left) is shown in solid red.

H.265 (Sullivan et al., 2012) in the ffmpeg implementation (Developers; VideoLAN, a;b). For our method and all baselines we use a GoP size of 5 frames and operate in the low-delay setting with only I-frames and P-frames. We trained our pipelines on TeslaV100 GPUs for approximately 300 GPU hours for each video. We show the average performance over all videos in Fig. 2. Our method is able to compete with mpeg4-2 at low bit rates, but it is still clearly behind H.264 and H.265. We do not compare to Chen et al. (2021) here because it is not a low-delay model.

Our performance over naive SIREN relies mainly on two aspects, quantization and frame-by-frame redundancies. Regarding the former, our quantization scheme compresses each model to around 10 bits per parameter with minimal degradation in performance. To gain some insights into our frame-by-frame scheme, we breakdown the rate-distortion performance per type of frame in Fig. 2. The red solid line indicates our average performance as in Fig. 2. This is averaged over I-frames (dot dashed dark blue) and P frames (dashed light blue). The I-frames are 2 - 5 dB higher in PSNR than the P-frames, depending on the model size, while being 20 times as expensive to compress (considering the combined rate of flow and residual, each of which is 1/40 size of the I-frame). We also consider the simpler codec without the residual component. These perform marginally worse than the full codecs.

## 5 CONCLUSION

We propose a video compression codec based on implicit neural representations. Our scheme compresses videos frame-by-frame, taking into account the similarity across frames, and is thus suitable for streaming applications. While not out-performing popular codecs, our method have the practical advantage of not requiring training set or decoder-side information, a practical advantage to most existing neural codecs.

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
