# OpenReview forum: "Implicit Neural Video Compression"
_ICLR.cc/2022/Workshop/DGM4HSD — ICLR 2022 DGM4HSD workshop Poster_

### Official Review · Reviewer_KuSz · 2022-03-15
**Perhaps a poster, main issue is clarity for the target audience**

**Rating:** 5
**Confidence:** 2

**Review:**

### Goal

The ultimate goal of this work is to prescribe a "codec", a method of compressing and de-compressing video sequences. Subgoals shared by most such codecs include: preserving high image quality; requiring low bitrate; have low decompression processing cost; and can be streamed (e.g., can start to watch the video before it has been decompressed in its entirety).

This work uses an implicit neural network strategy: each (I or P) frame is encoded as the quantized weights of an implicit neural network. Specialized subgoals/differentiators of relevance to this strategy include: no need to have a pretrained neural network shared beforehand by the involved parties; and no need for a separate training dataset.


### Description

In my current state of understanding, I personally could not replicate the work based on the manuscript, and it is not clear to me how much this situation would change if I were to fully process the relevant literature. That being said, given the 4 pages limit, I deem the manuscript to be adequately detailed to understand the important ideas and distinctions with other work: this is good enough for a workshop poster.


### Evaluation

Figures are provided as preliminary evaluation: Fig. 1 has examples of frames to be subjectively evaluated by the reader; Fig. 2(left) shows Peak Signal to Noise Ratio at different bitrates; and Fig. 2(right) breaks down the PSNR@rate for different frame types. The reported numbers are notably inferior to those of traditional codecs, but they are in the right ballpark so that one may hope that future work could catch up.

The authors note "We do not compare to Chen et al. (2021) here because it is not a low-delay model." That paper's NeRV model clearly beats the authors' IPF on the PSNR@rate metric, but I agree that NeRV not being a streaming algorithm is a point on IPF's side.

Future versions of the paper should consider reporting other metrics, such as Multi-Scale Structural Similarity Index Measure (MS-SSIM).


### Significance

I believe that the manuscript advances the state of understanding in the domain of video compression to a sufficiently extent to justify a workshop poster.

However, the fit for this specific workshop is not crystal clear. "Video" is explicitly listed among the structured modalities in the organizers' call for papers, but it is not clear to me whether each frame being represented as an overfitted generative model qualifies.

In the end, I believe that the real question is "would the kind of people attracted by the DGM4HSD workshop find this work interesting?". Personally, I learned a lot of interesting things while reviewing this manuscript, but it demanded a substantial amount of effort on my part to browse the relevant literature myself. This train of thoughts continues in the clarity section.


### Clarity

The guidelines to reviewers ask "Is it written in a way such that an interested reader with a background in machine learning and/or topology, but no special knowledge of the paper's subject, could understand and appreciate the paper's results?", to which my answer is "no". Section 2's first paragraph "Implicit neural representations" lists things that have been done with such implicit methods, but not what those methods *are* nor *do*. The same could be said of many other prerequisites of the manuscript.

The manuscript's overall grammar/quality/disclosure/etc look good to me, but it is just not targeted at an DGM4HSD workshop attendee (nor at myself).


### Recommendation

This is definitely not fit for an oral presentation at DGM4HSD. If there is no limit to the number of accepted posters and if the committee deem this contribution appropriate after reading my review, I would not object to it being accepted as a poster: the materials look sound and correct to me, the problem is clarity for the target audience.

To the authors: if this ends up being accepted as a poster, then I highly recommend you to take the time to add supporting pictures/examples/equations to help you explain to the attendees the fundamental prerequisites. Imagine yourself talking to someone that has a deep understanding of some subfield of machine learning, but whose sole exposure to "picture stuff" is a 10 years old CNN. I believe that this would highly increase the chance for you to have interesting discussions during the poster session, and/or that you take ideas/collaborations home from DGM4HSD.

---

### Decision · Program_Chairs · 2022-03-28

Accept (Poster)